# Mutation of the *Drosophila melanogaster* serotonin transporter *dSERT* impacts sleep, courtship, and feeding behaviors

Elizabeth M. Knapp[1], Andrea Kaiser[2], Rebecca C. Arnold[1], Maureen M. Sampson[1], Manuela Ruppert[2], Li Xu[2], Matthew I. Anderson[3], Shivan L. Bonanno[1], Henrike Scholz[2], Jeffrey M. Donlea[4], David E. Krantz[1]*

1 Department of Psychiatry, University of California, Los Angeles, California, United States of America, 2 Department of Biology, Institute of Zoology, Albertus-Magnus University of Cologne, Cologne, Germany, 3 Hamilton College, Clinton, New York, United States of America, 4 Department of Neurobiology, University of California, Los Angeles, California, United States of America

* dkrantz@ucla.edu

## Abstract

The Serotonin Transporter (SERT) regulates extracellular serotonin levels and is the target of most current drugs used to treat depression. The mechanisms by which inhibition of SERT activity influences behavior are poorly understood. To address this question in the model organism *Drosophila melanogaster*, we developed new loss of function mutations in *Drosophila SERT* (*dSERT*). Previous studies in both flies and mammals have implicated serotonin as an important neuromodulator of sleep, and our newly generated *dSERT* mutants show an increase in total sleep and altered sleep architecture that is mimicked by feeding the SSRI citalopram. Differences in daytime versus nighttime sleep architecture as well as genetic rescue experiments unexpectedly suggest that distinct serotonergic circuits may modulate daytime versus nighttime sleep. *dSERT* mutants also show defects in copulation and food intake, akin to the clinical side effects of SSRIs and consistent with the pleomorphic influence of serotonin on the behavior of *D. melanogaster*. Starvation did not overcome the sleep drive in the mutants and in male *dSERT* mutants, the drive to mate also failed to overcome sleep drive. *dSERT* may be used to further explore the mechanisms by which serotonin regulates sleep and its interplay with other complex behaviors.

## Author summary

Many medications used to treat depression and anxiety act by changing serotonin levels in the brain. Fruit flies also use serotonin and can be used as a model to study the brain. We have made a fly mutant for the serotonin transporter (SERT), which is the target of antidepressants in humans. The mutants sleep more, eat less, and have a decreased sex drive. These flies can be used to study the neuronal pathways by which serotonin regulates sleep, eating and sexual behaviors and may help us to understand the behavioral effects of antidepressants.

**Funding:** Funding for this work included R01 MH107390 (DEK), R01 MH114017 (DEK), a seed grant from the UCLA Depression Grand Challenge (DEK), R01NS105967 (JMD), an Early Career Development Award from the Sleep Research Society Foundation to (JMD), a Career Development Award from the Human Frontiers Science Program (CDA00026-2017-C) (JMD), a Grant of the Thyssen Foundation (HS) and TR 1340: Ingestive Behaviour: Homeostasis and Reward (HS). Additional support included a National Science Foundation GRFP (MMS), a UCLA Cota-Robles fellowship (MMS), F99 NS113454 (MMS) and F32 NS123014 (EMK). The funders had no role in study design, data collection and analysis, decision to publish, or preparation of the manuscript.

**Competing interests:** The authors have declared that no competing interests exist.

## Introduction

Sleep is essential for survival and is evolutionarily conserved from insects to mammals [1–5]. Both the amount and architecture of sleep can influence cognition and disruption of sleep in humans is linked to neurological and psychiatric disorders [6–8]. The neuromodulator serotonin (5-hydroxytryptamine, 5-HT) acts as a key regulator of sleep, and its involvement in sleep has been known for decades [9,10]. Previous studies in vertebrates have shown that serotonin signaling can promote wakefulness, while paradoxically, others demonstrate that serotonin is required for sleep induction and maintenance [11–15]. The circuits responsible for these effects and the cellular mechanisms by which serotonin regulates sleep remain unclear.

For over two decades, *D. melanogaster* has been utilized as a model system to study sleep [16,17] and serotonin [18–22] as well as other neuromodulators [23,24] have been shown to play significant roles in regulating sleep duration and quality. A role for serotonin in promoting sleep in *D. melanogaster* has been demonstrated in part by feeding the precursor 5-hydroxytryptophan (5-HTP) [18] or by utilizing mutants for the serotonin rate-limiting, synthetic enzyme tryptophan hydroxylase (TRH) [22]. Furthermore, of the five serotonin receptors expressed in *D. melanogaster*, mutations in three (*d5-HT1A*, *d5-HT2B* and *d5-HT7*) disrupt sleep behaviors, including total sleep amount, sleep rebound, and/or sleep architecture [18,20,22]. The powerful molecular genetic tools available in the fly have allowed some of the structures and cells required for sleep to be identified [18,22,25–32]. These tools have the potential to further dissect the mechanisms by which specific serotonergic circuits regulate sleep.

Serotonin modulates a variety of other behaviors in the fly including feeding [33–38] and sexual behavior [38–43]. It is possible that some of the circuitry underlying serotonin's effects on other behaviors is shared by pathways that regulate sleep. Alternatively, the serotonergic pathways for different behaviors may be hierarchical or otherwise conflict with one another. These relationships remain poorly understood.

The primary mechanism by which serotonin is cleared from the extracellular space in both mammals and flies is reuptake via the Serotonin Transporter (SERT) which localizes to the plasma membrane of serotonergic neurons [44]. Blockade of SERT activity increases the amount of serotonin available for neurotransmission. It is the primary target for most current treatments of depression and anxiety disorders, including the widely prescribed Selective Serotonergic Reuptake Inhibitors (SSRIs) [45]. In addition to their therapeutic effects, SSRIs can dramatically influence a variety of other physiological functions and behaviors such as eating, libido, and sleep [46–49]. Consistent with the complex relationship between serotonin and sleep, SSRIs often produce diverse and sometimes contradictory defects on sleep including insomnia, decreased REM sleep efficiency and daytime somnolence [11,49,50]. The specific circuits and mechanisms by which changes in SERT activity influence these behaviors remain unclear.

A relatively weak hypomorph of *dSERT* has been previously described [51]. The *dSERT* MiMIC insertion lies within the first intron just upstream of the first coding exon of the gene and reduces *dSERT* mRNA expression by ~50% [51]. Its potential effects on sleep were not reported, and in general, it can be difficult to make firm conclusions about mutations that are not null or at least strong hypomorphs.

In this study, we have used P-element excision to generate new mutant alleles and find that *dSERT* is required for regulating both sleep amount and architecture. Our work further elucidates how the increased sleep drive exhibited in these *dSERT* mutants is affected in the context of other critical behaviors that are regulated by serotonin signaling including mating and feeding. Our data also suggest that *dSERT's* role in daytime versus nighttime sleep may be modulated via distinct serotonergic circuits.

## Results

### Disruption of *dSERT* increases sleep in *D. melanogaster*

To generate new mutant alleles of *dSERT*, we used imprecise excision of a transposable element in the line $XP^{d04388}$ (BDSC #85438). To ensure a consistent genetic background, $XP^{d04388}$ was first outcrossed into $w^{1118}$, and $w^{1118}$ served as the primary control for our initial assays. The proximal end of the P element in $XP^{d04388}$ is 514 bp 5' of the predicted transcriptional start site of *dSERT* (Fig 1A). We screened for loss of the *white* minigene in the P element and recovered two imprecise excision alleles of 1121bp and 1178bp which we designate as $dSERT^{10}$ and $dSERT^{16}$ respectively (Fig 1A). Both $dSERT^{10}$ and $dSERT^{16}$ delete the first non-coding exon and the first intron of *dSERT*. We identified two additional lines as controls; $dSERT^{1}$ contains 41 additional bases that are remnants of the former P-element insertion but does not otherwise disrupt the *dSERT* gene and $dSERT^{4}$ with no detectable genomic alterations at the former P-element insertion site.

To determine whether expression of *dSERT* was disrupted in $dSERT^{10}$ and $dSERT^{16}$, we first used qRT-PCR to quantify mRNA expression, using $w^{1118}$ as a control. The *dSERT* transcript was not detectable in either of the mutant alleles (Fig 1B). By contrast, the previously published mutant allele generated by insertion of a MiMIC cassette was reported to retain ~50% expression relative to wild type [51]. To confirm these results and determine whether dSERT protein was similarly reduced, we performed western blots using a previously validated antibody to dSERT [52]. Compared to $w^{1118}$ controls, both $dSERT^{10}$ and $dSERT^{16}$ show a decrease in dSERT protein expression (Fig 1C). It is unclear whether a faint band present in the western blots represents residual protein or non-specific background, and the intact coding sequence suggests that they may not be null alleles. Nonetheless, since they appeared to represent relatively strong hypomorphs with undetectable mRNA levels we preceded with our behavioral analyses.

We analyzed sleep in the *dSERT* mutants and found that both $dSERT^{10}$ and $dSERT^{16}$ mutants have dramatically increased sleep compared to $w^{1118}$ controls (Fig 1D). To confirm the increased sleep phenotypes are the result of *dSERT* disruption and not from other spurious changes to the genetic background, transheterozygous *dSERT* mutants ($dSERT^{10}/dSERT^{16}$) were assayed and shown to exhibit a significant increase in total sleep compared to controls (S1A–S1B Fig). Consistent with these findings, $dSERT^{16}$ homozygotes also demonstrated a significant increase in total sleep when compared to heterozygous $dSERT^{16}$ flies (S1C–S1D Fig). To determine how genetic background may contribute to the behavior we observed, sleep in $dSERT^{16}$ was also compared to the revertant $dSERT^{4}$ control, which was obtained in the same screen as the imprecise excisions and shares the same genetic background. We also assayed sleep in the parental P-element line *P{XP}d04388* (S1E–S1F Fig). We did not detect any difference in sleep between $w^{1118}$ and $dSERT^{4}$ and only a slight difference between $w^{1118}$ and the parent line, with the parent showing a slight decrease in sleep compared to $w^{1118}$. Although it is impossible to completely rule out subtle effects caused by genetic background, these results demonstrate disruption of *dSERT* is primarily responsible for the increase in overall sleep and sleep drive we observe in the *dSERT* mutants.

### Loss of *dSERT* significantly increases sleep in daytime and nighttime and alters sleep architecture

The sleep phenotype we observed was stronger in $dSERT^{16}$ compared to $dSERT^{10}$, and we therefore focused on $dSERT^{16}$ for further analysis. To verify that the phenotype was not due to a bias in the time spent on one end of the testing tube (e.g., near the food source) or to limited

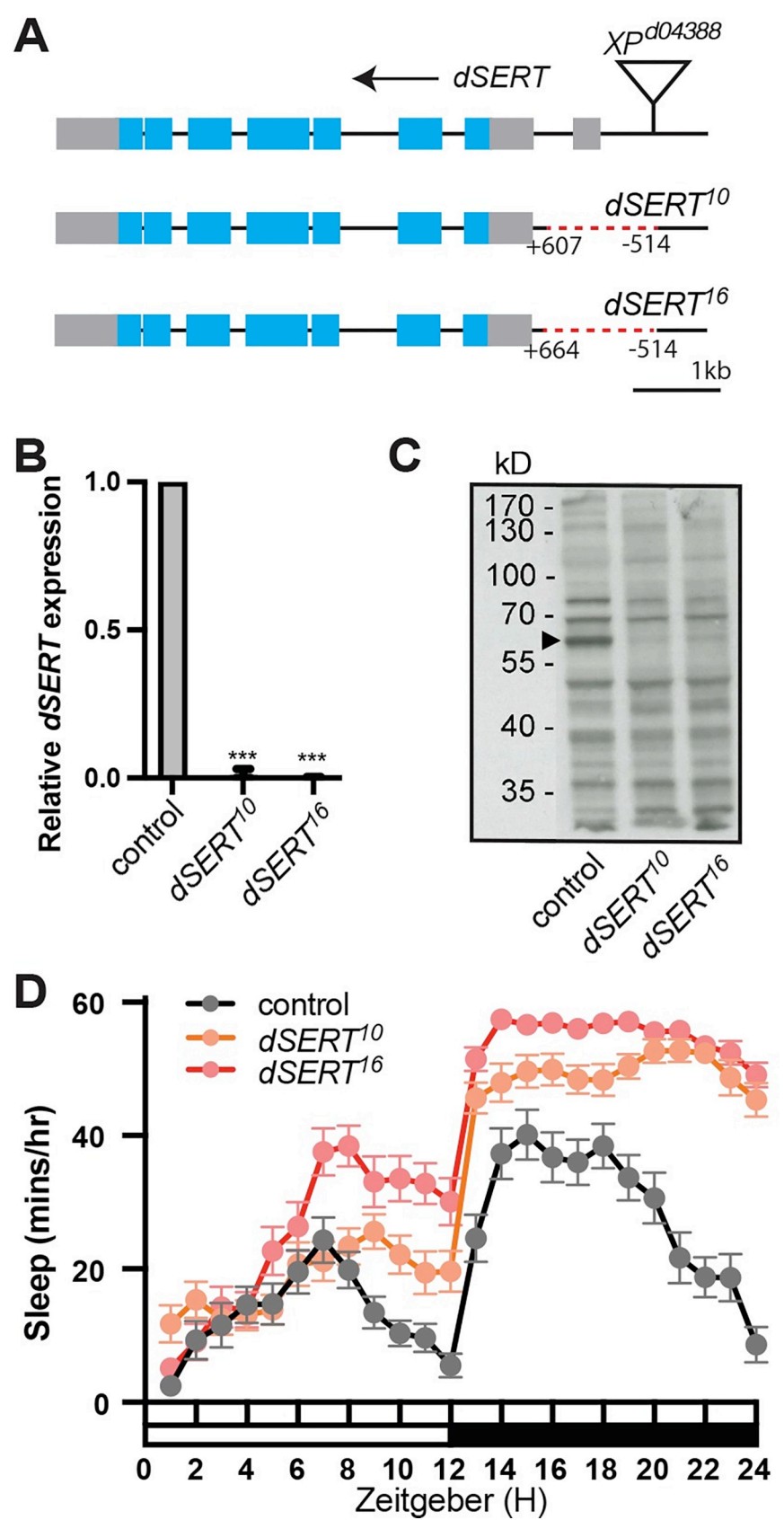

**Fig 1. *dSERT* imprecise excision alleles.** (A) Schematic of *dSERT* gene: grey and blue boxes indicate non-coding exons and coding exons respectively. Deletions of the *dSERT[10]* and *dSERT[16]* flies are indicated with red dotted lines. (B) qRT-PCR was performed on cDNA synthesized from whole, male flies with the ubiquitously expressed ribosomal RNA *RplP0* used as a reference. Gene expression of *dSERT* in the mutants was normalized to controls with +1 representing expression in *w[1118]*. *dSERT* transcript levels were significantly downregulated in *dSERT[10]* (0.01±0.02) and *dSERT[16]* (0.001±0.003) compared to controls. Error bars represent SD. P***≤0.001, Student's t-test. (C) Western blot analysis shows a band representing dSERT at ~65kD that is reduced in intensity in both mutants. Actin was used as a loading control and shows no difference across genotypes (not shown). (D) Hourly sleep traces in *w[1118]* (black) *dSERT[10]* (orange) and *dSERT[16]* (red) homozygotes. (Sleep trace shows mean ± SEM, n = 31–32 flies per group).

movements that do not carry the flies past the tube's midline, we repeated the assays with monitors that use 4 separate beams to detect movement [53]. Since multiple sites within the tube are simultaneously assessed for activity, multibeam monitors provide a more sensitive assay, and a more conservative assessment of sleep. Our results show that in both single-beam and multibeam monitors *dSERT[16]* mutants exhibit a significant increase in total sleep compared to controls (Fig 2A).

Previous studies have shown that stimulation of aminergic pathways can increase grooming and alter other sensorimotor behaviors that might affect sleep [54–56]. We therefore tested whether grooming or negative geotaxis were altered in the *dSERT* mutants. We did not detect any differences from controls in either grooming rate (S2A Fig) or performance of negative geotaxis in *dSERT[16]* flies (S2B Fig). Overall, these findings indicate that the behavioral change we detect represents increased sleep in *dSERT* mutants rather than an artifact caused by changes in other amine-associated behaviors.

More specific analysis of both daytime and nighttime sleep behavior shows that *dSERT[16]* flies have increased sleep during the daytime (Fig 2B). This is further characterized by an increase in daytime bout frequency compared to controls (Fig 2C). In addition, *dSERT[16]* mutants displayed reduced latency to sleep at night (Fig 2D). The analysis of nighttime sleep behavior revealed that *dSERT[16]* mutants similarly exhibit dramatically increased nighttime sleep (Fig 2E) but in contrast to their daytime behavior, we observed a significant decrease in nighttime bout frequency (Fig 2F). The probability of transitioning from an awake to a sleep state, P(Doze), and the probability of transition from a sleep to an awake state, P(Wake), were calculated to further analyze the changes in sleep drive and arousal in the *dSERT* mutants [57]. Compared to controls, both *dSERT* mutants show a significant increase in P(Doze) (S1H Fig) and a reduction in P(Wake) (S1G Fig). Together, these data indicate that disruption of *dSERT* causes a significant increase in sleep and alters sleep architecture in both the day and night periods.

To further confirm that increased sleep is a result of disrupted dSERT function, we tested whether feeding adult flies an SSRI to block dSERT activity would impact sleep. *w[1118]* flies fed the SSRI citalopram showed a dose dependent increase in total sleep (Fig 3A). Consistent with our findings with the *dSERT* mutants, both daytime sleep (Fig 3C) and nighttime sleep (Fig 3D) were significantly increased in wildtype flies fed citalopram. We do not detect any further increase in sleep in *dSERT[16]* mutants fed citalopram (Fig 3B) supporting the idea that citalopram's effect on sleep is mediated by inhibition of dSERT rather than a spurious, off-target effect. In addition, these findings indicate that an increase in adult sleep can be caused by disrupting dSERT only in the adult and not during development. However, further experiments will be needed to determine whether additional, and possibly more subtle developmental defects may occur in the mutants.

## *dSERT[16]* mutants exhibit rhythmic circadian behaviors

In addition to sleep, serotonergic pathways in mammals have been implicated in the regulation of circadian rhythmicity [58]. To investigate a possible role for dSERT activity in sustaining

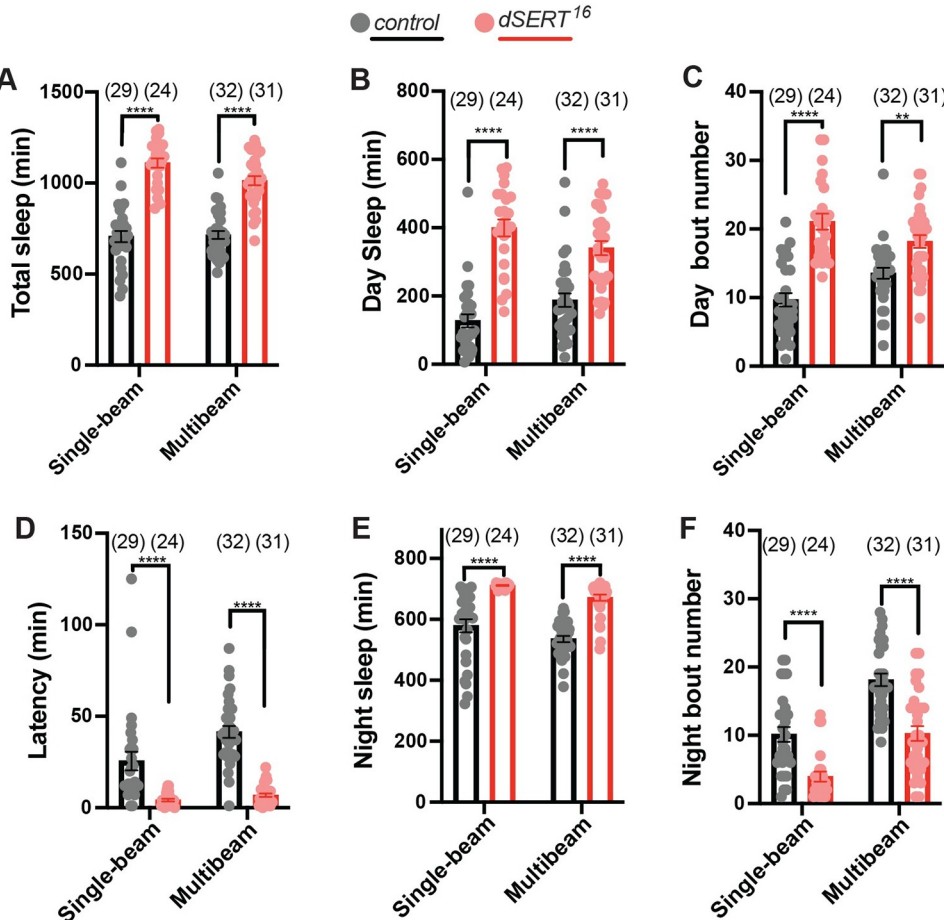

**Fig 2. *dSERT^16* mutants exhibit increased sleep behavior and changes in sleep architecture.** Sleep was separately recorded with single-beam and multibeam monitors. (A) Quantification of total sleep in *w^1118* controls (grey) and *dSERT^16* homozygous mutants (red). Analysis of daytime sleep (B) and daytime bout frequency (C). (D) Latency after light-off was significantly decreased in *dSERT^16* mutants. Quantification of nighttime sleep (E) and nighttime bout frequency (F). Graphs show individual datapoints and group means ± SEM, two-way ANOVA with Tukey post-hoc test, (p≤0.0021**, p≤0.0001****). Comparison of single beam versus multibeam data showed no significant differences for panels A, B, E; p<0.0332 * for panel C; p<0.0021 ** for panel D; and p<0.0002 *** for panel F.

circadian rhythms, we analyzed circadian locomotor behaviors in *dSERT^16* mutants. Both control and *dSERT^16* mutants exhibited robust locomotor rhythms with bimodal activity peaks in 12h/12h light/dark (LD) cycles (Fig 4A and 4B) and their rhythmic behaviors persisted in free-running constant darkness (DD) cycles (Fig 4E and 4F). The average activity of *dSERT^16* flies while awake (activity counts/time awake) did not differ from control flies in LD (Fig 4C) nor DD (Fig 4G). In addition, *dSERT^16* mutants did not exhibit a significant decrease in either morning or evening anticipation behaviors for either LD (Fig 4D) or DD (Fig 4H) cycles. Periodogram analysis confirmed that *dSERT^16* mutants behave indistinguishably from wildtype controls (Fig 4I and 4J) and quantification of free-running circadian behaviors demonstrates that all of the *dSERT^16* flies (tau = 23.6 ±0.04) were rhythmic (Fig 4K) similar to control flies (tau = 23.5±0.01). In addition, the circadian periods of *dSERT^16* mutants did not detectably differ from controls Fig 4L). In summary, we find that loss of *dSER*T does not detectably disrupt circadian rhythmicity. These data are consistent with previous work showing the feeding of either the serotonin precursor 5-HTP or the SSRI fluoxetine (Prozac), or overexpression of *d5-HT1B* did not detectably alter circadian rhythms in free-running conditions [19].

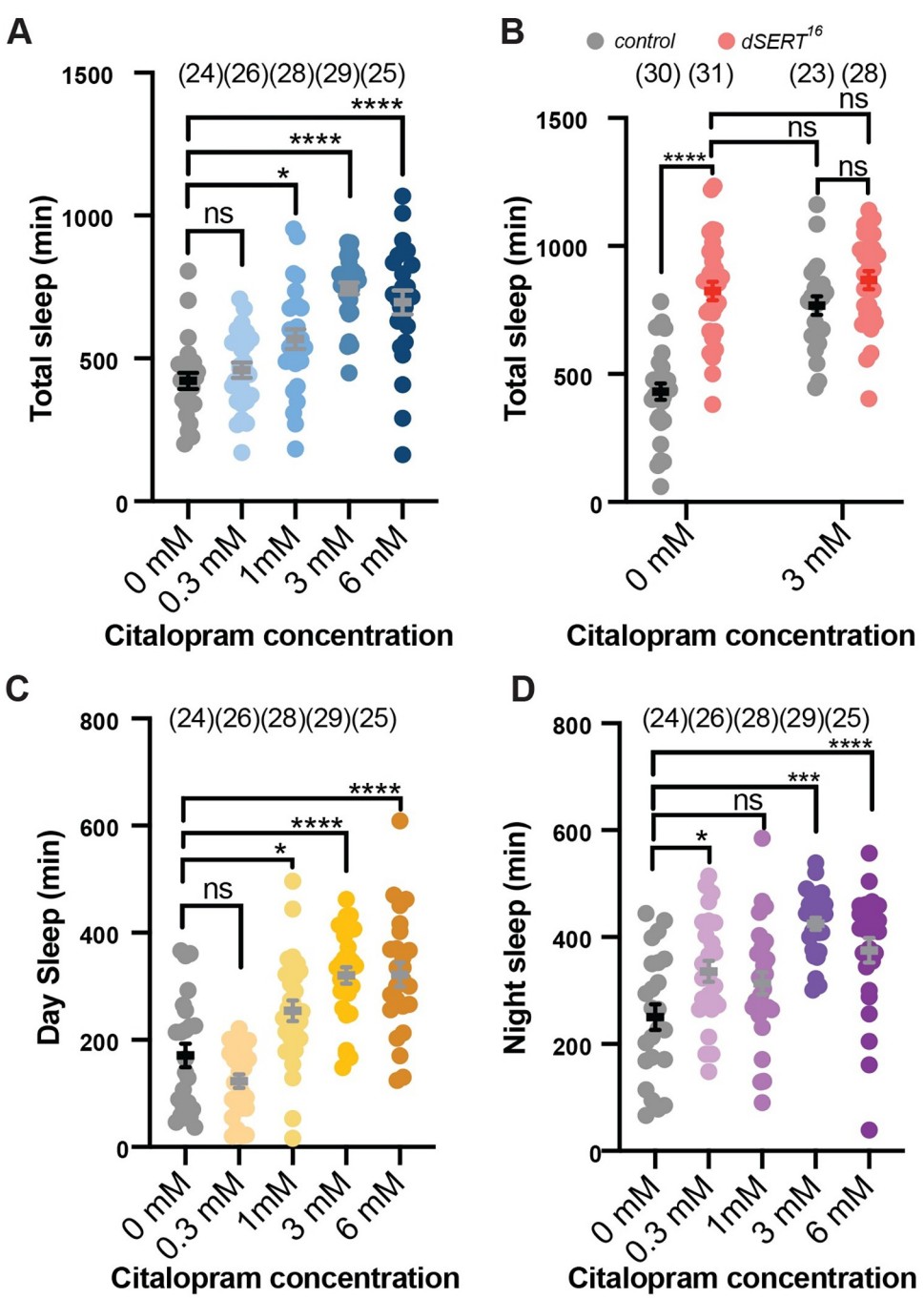

**Fig 3. Wildtype flies fed SSRIs exhibit increased sleep.** (A) Quantification of total sleep in $w^{1118}$ control flies fed the indicated concentrations of citalopram (0- 6mM). (B) Total sleep in $w^{1118}$ controls (grey) and $dSERT^{16}$ mutants (red) fed vehicle (0 mM) or citalopram (3 mM). Analysis of daytime sleep (C) versus nighttime sleep (D) for the flies tested in panel A. Graphs show individual datapoints and group means ± SEM. A,C,D one-way ANOVA; B two-way ANOVA, with Tukey post-hoc test ($p \leq 0.0332^*$, $p \leq 0.0002^{***}$, $p \leq 0.0001^{****}$).

## Sex dependent effects in *dSERT* mutants

All of our initial experiments analyzed sleep in mated females. Males and females exhibit different sleep patterns, with males exhibiting a more pronounced daytime siesta [59,60]. In

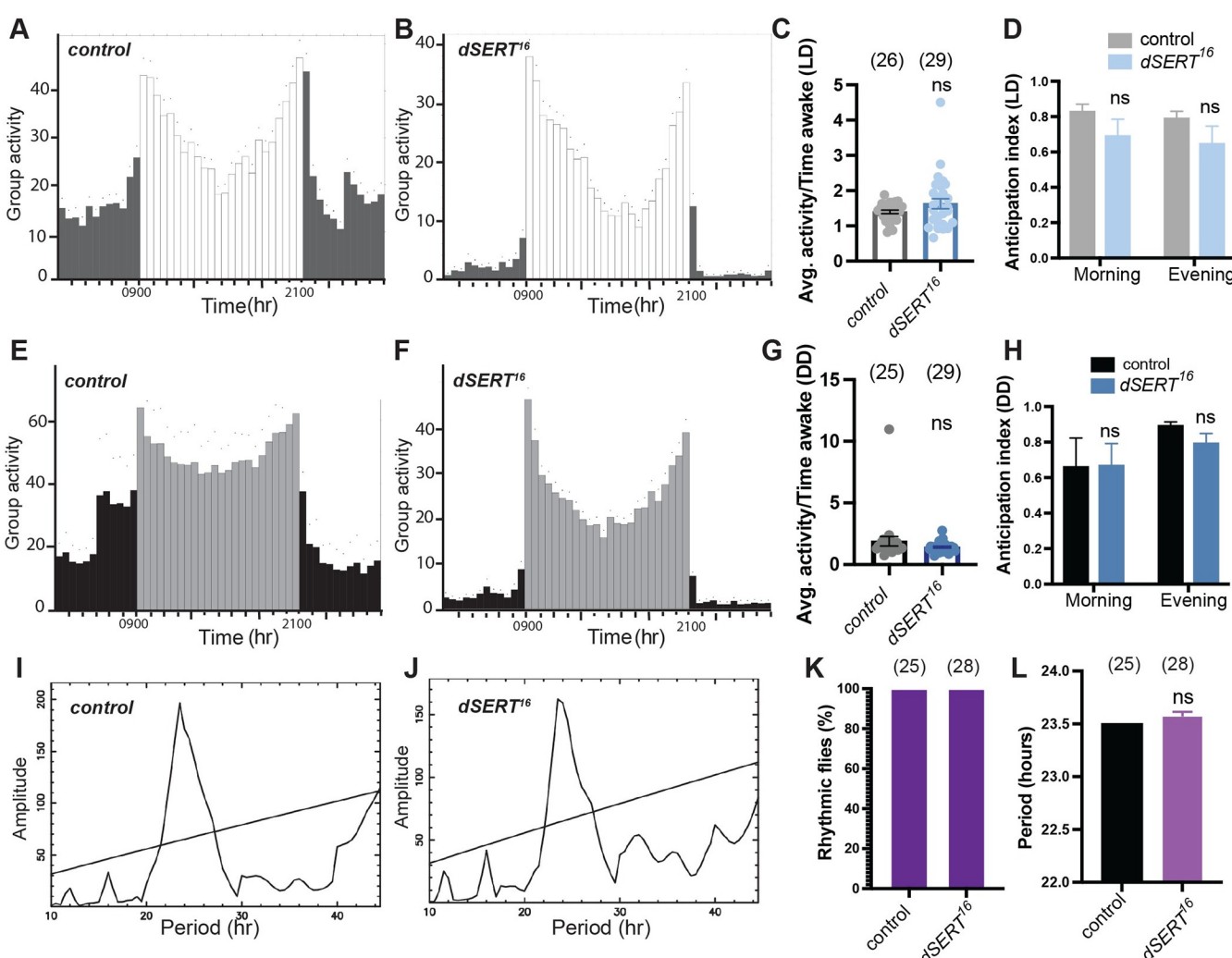

**Fig 4. *dSERT¹⁶* mutants exhibit rhythmic circadian behaviors.** (A-B) 12h:12h LD and (E-F) 12h:12h DD locomotor activity group analysis for *w¹¹¹⁸* (A,E) and *dSERT¹⁶* (B,F) flies. Histograms represent the distribution of activity through 24 h, averaged over three days, number of flies indicated in C, D, G, H. Lighter and darker bars indicate day and night phases, respectively with 0900 lights-on and 2100 lights-off and each column representing the mean, binned activity for 0.5 hr, n = 25–29 flies. Small dots above each column indicate the SEM. (C,G) The average activity while awake in *dSERT¹⁶* mutants is no different than controls in both LD (C) and DD (G) cycles, individual datapoints and group means ± SEM. Student's t-test, two-tailed unpaired. (D, H) Calculation of morning and evening anticipation indexes for *dSERT¹⁶* and control flies in LD (D) and DD (H) cycles. (I-J) Representative periodograms derived from activity records of individual *w¹¹¹⁸* (I) and *dSERT¹⁶* (J) flies in constant darkness. (K) Percentage of flies with detectable rhythmicity was calculated for controls and *dSERT¹⁶*. (L) Circadian periods in DD were averaged from rhythmic flies per each genotype, error bars indicate SEM.

addition, the post-mating response in females includes a decrease in daytime sleep [61]. We therefore compared the effects of *dSERT* on males, mated females, and virgin females (S3 Fig). Similar to mated females, total sleep (S3A Fig), daytime sleep (S3B Fig), and nighttime sleep (S3E Fig) were significantly increased in *dSERT* mutant males and virgin females compared to controls. For all three groups (mated females, male and virgin females) we also detect a reduction in latency in the *dSERT* mutant compared to controls, and a consistent change in nighttime sleep architecture, observed as a decrease in sleep bout frequency. Interestingly, in contrast to nighttime sleep architecture, the effects of *dSERT¹⁶* on *daytime* architecture differed between the three groups; mated *dSERT* females showed an increase in daytime bout frequency compared to controls that was not observed in either males or virgin females (S3C Fig).

## Loss of *dSERT* disrupts courtship and copulation behaviors

The difference between mated and virgin females in daytime sleep architecture suggested the possibility that *dSERT* could influence other sex-dependent behaviors. We therefore investigated whether *dSERT*[16] mutants might show defects in courtship and/or copulation and how this would interact with sleep. Previous work demonstrated that pairing male and female flies together overnight dramatically suppresses sleep due to increased courtship activity [62–64]. Consistent with these findings, co-housing pairs of wildtype male and female flies together resulted in a significant decrease in nighttime sleep compared to sleep recorded from isolated male or female flies, or female-female pairs (Fig 5A). In contrast to the controls, we did not detect a decrease in sleep when male and female *dSERT*[16] mutants were paired compared to single mutant flies or female-female pairs (Fig 5A). The failure of male-female pairing to decrease sleep in *dSERT*[16] mutants suggest that increased sleep drive outweighs copulation drive in the mutant. These data also suggested the possibility that loss of *dSERT* could decrease the drive for mating activity, but because the pairs were homotypically mutant or WT these data did not distinguish between a male and/or female phenotype for *dSERT*.

To determine whether reduced mating activity in *dSERT*[16] mutants was a male and or female phenotype, we paired mutant males or females with wild type flies of the opposite sex. Pairing *dSERT*[16] mutant females with control males significantly reduces sleep compared to *dSERT*[16] male-female coupling. By contrast, we did not detect a statistically significant reduction in sleep when *dSERT*[16] males were coupled with wild type control females (Fig 5B). These data suggest that *dSERT*[16] males, but not females, show defects in sexual behavior.

To more directly test whether *dSERT* mutant males and/or females would show a sex- phenotype, we measured copulation rates (Fig 5C). We did not observe a significant difference in copulation success in *dSERT*[16] females compared to control females when paired with wildtype males. By contrast, when *dSERT*[16] males were paired with wildtype females, none of the pairs copulated within a standard period (one hour), in marked contrast to behavior of control males, of which 95% copulated with WT flies within the same time frame (Fig 5C). Since copulation and courtship might be differentially effected in *dSERT* males, we also assayed courtship behavior. *dSERT*[16] males exhibited significant defects in courtship behavior including increased latencies to initiate orientation (Fig 5D) and wing song (Fig 5E).

As a further test of male sexual behavior, we measured reproductive output in wildtype females paired for 2 days with *dSERT*[16] males. Wild type females paired with *dSERT*[16] males showed a reduction in the rate of egg laying (Fig 5F) as well as an increase in the proportion of unfertilized eggs (Fig 5G). These results indicate that while *dSERT*[16] males fail to copulate within one hour, over an increased time-period mating can occur, albeit with reduced fecundity compared to controls.

## Loss of *dSERT* reduces food intake

Previous work has demonstrated that starvation as well as mating induces sleep loss in *D. melanogaster* presumably because starvation increases the drive to search for food during periods that are normally occupied by sleep [65]. We therefore tested whether starvation was sufficient to suppress sleep in *dSERT*[16] mutants. Consistent with previous studies, control flies dramatically suppressed their sleep when starved on agar medium compared to their sleep on baseline or recovery days when housed with standard food (Fig 6A). Surprisingly, starvation did not have a significant effect on sleep in the *dSERT*[16] mutants (Fig 6A). We directly tested feeding in the *dSERT* mutants and found that after 24 hours of starvation, food uptake in *dSERT*[16] was significantly decreased compared to controls (Fig 6B–6D). Together these results indicate that loss of *dSERT* not only enhances sleep but also reduces food intake.

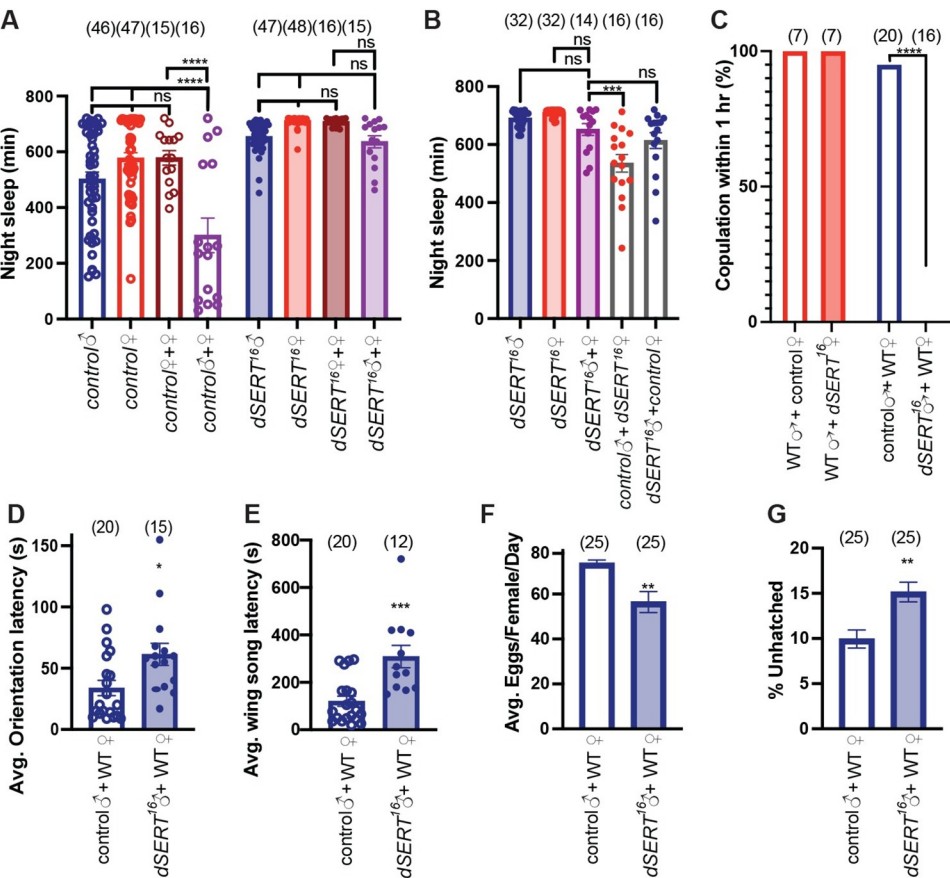

**Fig 5. *dSERT^16* mutants exhibit defects in courtship and copulation.** (A) Quantification of nighttime sleep for males (blue) and females (red) individually housed in DAMs tubes for both control *w^1118* (open bars/circles) and *dSERT^16* (shaded bars/circles). Following 2 days of individual housing, male-female (purple) or female-female (dark red/burgundy) were paired together in DAMs tubes and nighttime sleep was quantified for both *w^1118* (open bar/circles) and *dSERT^16* (shaded bar/circles). (B) Quantification of nighttime sleep for *dSERT^16* males (shaded blue) and females (shaded red) individually housed or in male-female pairs (shaded purple) and for mixed co-housing of control males with *dSERT^16* females (open grey bar with red circles) or *dSERT^16* males with control females (open grey bar with blue circles). (C-G) Mating pairs made up of either wildtype (WT, Canton-S) males (red) or WT females (blue) with *w^1118* controls (open bars) or *dSERT^16* mutants (shaded bars). (C) Percentage of pairs that copulated within 1 hr. Quantification of average latency to orientation (D), and wing song (E). Quantification of egg laying (F) and percentage of eggs that failed to hatch 24 hours after being laid (G). All graphs (except C) show means ± SEM. A-B: one-way ANOVA, with Tukey post-hoc test (p≤0.0002***, p≤0.0001****); C, Fisher's exact test (P****≤0.0001); D-G, two-tailed unpaired t test (p≤0.0332*, p≤0.0021**, p≤0.0002***).

The decreased effect of starvation on sleep in *dSERT* mutants could result from an increased resistance to starvation or perhaps dehydration. We assayed starvation resistance by housing flies on an agar substrate and found that *dSERT* mutants exhibit a significantly decreased probability of survival compared to controls (Fig 6E), with the *dSERT* mutants reaching a 50% mortality rate approximately 15% earlier than the controls. To test the effects of dehydration/desiccation we housed flies in empty vials (devoid of food and water). In contrast to the effects of starvation, we found that *dSERT* mutants exhibit a slight increase in resistance to desiccation and survived ~9% longer than controls (Fig 6F).

The lack of sleep reduction seen in *dSERT* mutants when starved (Fig 6A) or co-housed with the opposite sex (Fig 5A) could also be due to a higher threshold for arousal. We compared the arousability of sleeping flies for *dSERT^16* mutants and controls by exposing them to

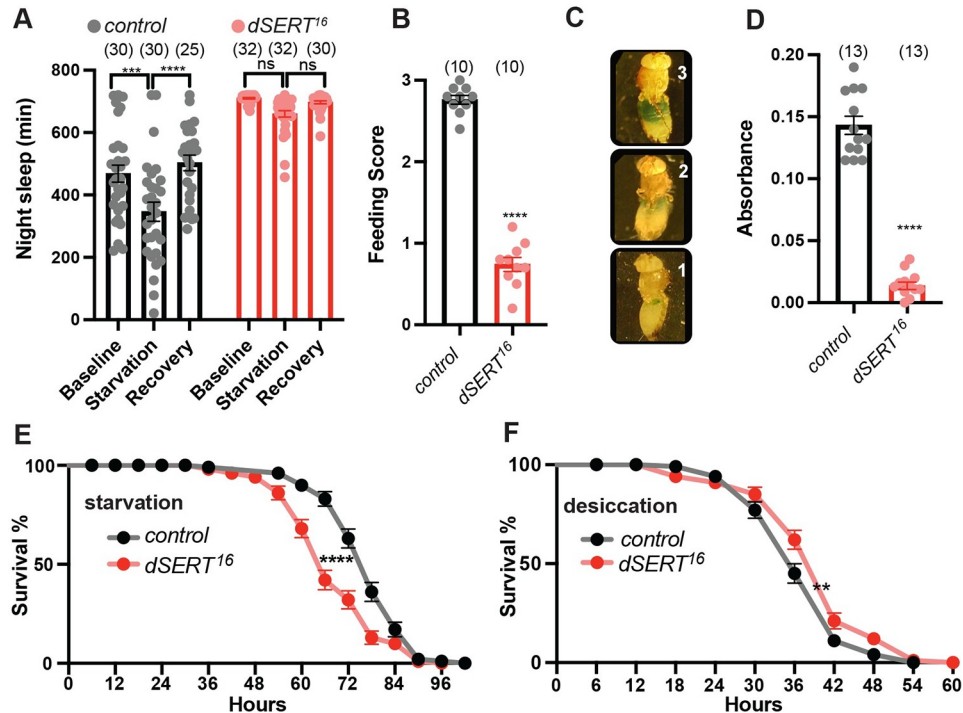

**Fig 6. *dSERT*[16] mutants exhibit defects in feeding.** (A) Quantification of nighttime sleep for wild-type *w*[1118] (grey) and *dSERT*[16] (red) over a 3-day period that began 1 day after initial loading into DAMs tubes. Baseline (Day 1): flies kept on standard food. Starvation (Day 2): flies transferred to agar for food deprivation. Recovery (Day 3): flies transferred to fresh food. (B-D) Food uptake is significantly reduced in starved *dSERT*[16] mutants (red) compared to controls (grey). (B) Average feeding score was calculated by visual inspection of abdomen dye levels (n = 10 groups of 10 flies). (C) Representative images depict visual feeding score used to assay food uptake. (D) Quantification via spectrophotometry of ingested blue dye (n = 13 groups of 10 flies). Fly survival rate under starvation stress (E) or desiccation stress (F). *dSERT* mutants (red) show decreased survival under starvation stress (E) and increased survival when compared to controls (black) under desiccation stress (F) (n = 100 flies for each genotype and experiment). Graphs (A,B,D) show individual datapoints and all show group means ± SEM. A, two-way ANOVA, with Tukey post-hoc test (p≤0.0002***, p≤0.0001****; B-D: two-tailed unpaired t test (p≤0.0001****); E-F, Log-rank Mantel-Cox test, (p≤0.0021**, p≤0.0001****).

5 seconds of mechanical vibration every hour during both the daytime and nighttime. We did not detect any differences in arousability between control and *dSER*T mutants when they were mechanically stimulated during the day using either stimulus (S4A Fig) or during the night using the stronger stimulus (1.0 g). By contrast, *dSERT* mutants exhibited a significant decrease in arousability compared to controls when exposed to a less intense vibrational stimulus (0.5 g) (S4B Fig). These data suggest that at night, *dSERT* mutants have a somewhat elevated arousal threshold, and an increased intensity of sleep. This decrease in arousability may contribute to the lack of wakefulness seen during starvation and co-housing conditions.

## Transgenic expression of *dSERT* in serotonergic neurons is sufficient to rescue nighttime and/or daytime sleep defects

To further confirm that defects seen in *dSERT*[16] mutants are due to a loss of *dSERT* we used "genetic rescue" and expressed a wild type *dSERT* transgene in the *dSERT*[16] background, focusing primarily on defects in sleep amount. Although sleep in the fly has been generally approached as a single behavior, some genes and environmental factors can preferentially affect one phase [66–69]. Regulatory sequences from the gene that encodes the rate limiting

enzyme for serotonin biosynthesis *tryptophan hydroxylase* (*TRH*) have been utilized to generate multiple Gal4 lines that target broad populations of serotonergic neurons [22,70–73]. We first used the broad serotonergic driver designated as *"TPH-Gal4"* [71] to express *UAS-dSERT*. Although expression of *UAS-dSERT* using *TPH-Gal4* did not fully reverse the increase in total sleep across the entire 24-LD cycle (Fig 7A) expression of *dSERT* using *TPH-Gal4* was sufficient to restore the nighttime sleeping pattern to near control levels (Fig 7C). By contrast, day sleep was not altered in the *dSERT^16* mutants by *TPH-Gal4>UAS-dSERT* (Fig 7B). To extend our analysis we tested another broad serotonergic driver, *TRH-Gal4* [70]. While both *TRH-Gal4* and *TPH-Gal4* include fragments of the *Tryptophan Hydroxylase* gene, they have been reported to exhibit differences in expression including relatively lower expression for *TRH-Gal4* in processes that innervate the mushroom bodies [38] (see also S5 Fig). Similar to *TPH-GAL4*, total sleep levels across the entire 24-hour day were not fully rescued using *TRH-Gal4* to express *UAS-dSERT* in the *dSERT^16* background (Fig 7D). However, in contrast to *TPH-GAL4*, *TRH-Gal4* was sufficient to restore daytime sleep levels to that of controls (Fig 7E). Moreover, unlike *TPH-Gal4*, *TRH-Gal4* did not rescue nighttime sleep levels (Fig 7F). We next tested whether combining the serotonergic drivers could fully restore sleep levels across the 24-hour period. Our results show that genetic rescue using both *TPH-Gal4* and *TRH-Gal4* to express *UAS-dSERT* in the *dSERT* mutant is sufficient to restore sleep levels to that of the controls in both the daytime and nighttime (Fig 7G–7I). Overall, these results demonstrate that the sleep defects in *dSERT^16* mutants can be genetically rescued, confirming that this phenotype is due to disruption of the *dSERT* gene. In addition, they suggest that distinct serotonergic circuits may regulate day and night sleep.

## Discussion

We find that dSERT functions as a modulator of sleep, with loss of *dSERT* causing a robust increase in both daytime and nighttime sleep. Given that dSERT acts as the primary mechanism by which serotonin is cleared from the extracellular space, it is likely that *dSERT* mutants have an increase in the amount of serotonin available for neurotransmission. Their behavior is consistent with phenotypes caused by other changes in serotonergic signaling: feeding the serotonin precursor 5-HTP or overexpression of the rate-limiting enzyme *TRH* in dopaminergic and serotonergic cells [18] (using *Ddc-Gal4*) increases total sleep, similar to *dSERT* mutants. Conversely, a decrease in total sleep duration is exhibited by *TRH* mutants [22].

A role for serotonin has also been suggested to contribute to male courtship and mating based in part on studies of the gene *fruitless* (*fru*) [41,74]. Male specific Fru^M-positive abdominal-ganglionic neurons that innervate the reproductive organs have been shown to be serotonergic and *fru* mutants demonstrate defects in courting and mating events [41,74]. Furthermore, activation of serotonergic neurons causes defects in male copulation behavior [38]. Conversely, in females another study showed that knockout of *TRH* or silencing serotonergic neurons causes a reduction in virgin receptivity [43]. Our data are consistent with previous findings on the effects of serotonin on male sexual behavior [38]. In females, the decrease in receptivity was observed in flies with reduced serotonergic signaling [43], whereas loss of *dSERT* would predicted to increase extracellular serotonin and thereby increase serotonergic signaling. It is conceivable that *dSERT* females could more receptive than WT but we have not yet tested this possibility.

Previous studies have also addressed the complex relationship between increased serotonin signaling and feeding behavior in *D. melanogaster*. The use of either *TPH-Gal4* or *TRH-Gal4* to thermogenetically depolarize a broad population of serotonergic cells can significantly decrease food intake [33,38]. By contrast, the activation of a specific subset of serotonergic

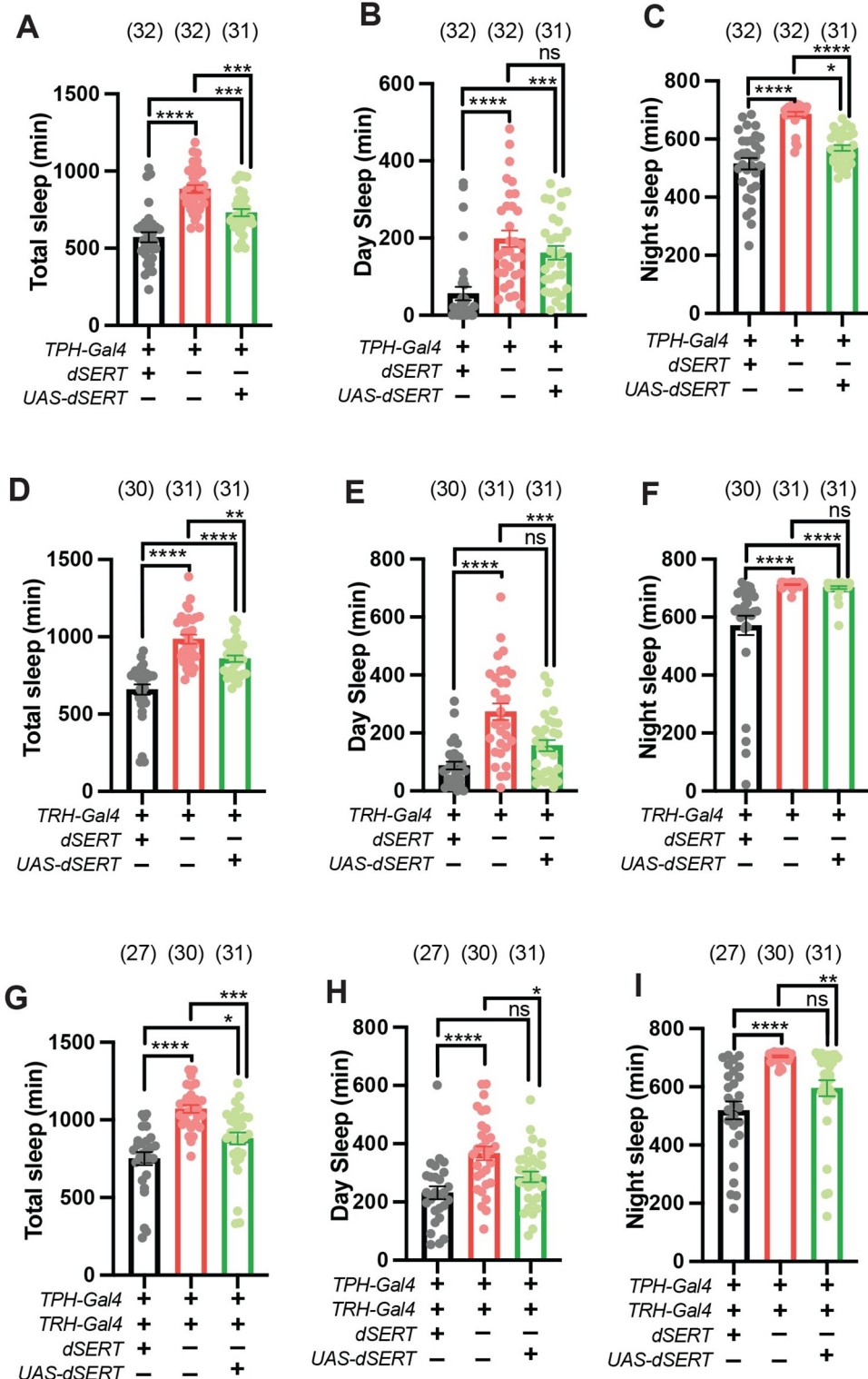

**Fig 7. Transgenic expression of *dSERT* with both "*TPH*"-*Gal4* and *TRH-Gal4* is sufficient to rescue increased sleep in *dSERT^16* mutants.** (A-C) Quantification of total sleep (A), daytime sleep (B) and nighttime sleep (C) in control *w^1118^;TPH-Gal4* (grey), *w^1118^;TPH-Gal4, dSERT^16* (red), and *w^1118^;TPH-Gal4, dSERT^16*; *UAS-dSERT* (green) flies. (D-F) Quantification of total sleep (D), daytime sleep (E), and nighttime sleep (F) in *w^1118^;TRH-Gal4* (grey), *w^1118^;dSERT^16*; *TRH-Gal4* (red), and *w^1118^;dSERT^16*; *TRH-Gal4/UAS-dSERT* (green) flies. (G-I) Quantification of total

sleep (G), daytime sleep (H), and nighttime sleep (I) in $w^{1118}$; *TPH-Gal4; TRH-Gal4* (grey), $w^{1118}$;*TPH-Gal4, dSERT*$^{16}$; *TRH-Gal4* (red), and $w^{1118}$;*TPH-Gal4, dSERT*$^{16}$; *TRH-Gal4/ UAS-dSERT* (green) flies. Graphs show individual datapoints and group means ± SEM. one-way ANOVA, with Tukey post-hoc test (p≤0.0332*, p≤0.0021**, p≤0.0002***, p≤0.0001****).

cells drives a contradictory behavior and induces feeding in sated flies [33]. The finding that broad activation of serotonergic cells reduces feeding is consistent with our data showing that globally expressed *dSERT* mutants exhibit a significant decrease in feeding behavior. It will be interesting to test whether cell-specific knock-down of *dSERT* could mimic the effects of activating a subset of serotonergic neurons [33,38].

Serotonin has also been implicated in the physiological functions of both the gut and the Malpighian tubules, the fly equivalent of mammalian kidneys [35,75–78]. For example, exogenous injection of serotonin into either the brain or the abdomen alters crop contractility [75]. These effects could potentially play a role in the differences in feeding behavior we observe in *dSERT* mutants, their sensitivity to starvation and/or their modest resistance to desiccation. Similar to *dSERT*, knockdown of either *5-HT1A* [37] or *5-HT1B* [35] can decrease resistance to starvation but increase resistance to desiccation. Further genetic experiments will be needed to determine whether these effects occur via independent pathways. We note that *5-HT1A* and *5-HT1B* are expressed in serotonergic neurons as well as post-synaptic neurons [79] and thus may function as autoreceptors. If so, knockdown of the receptors in pre- versus post-synaptic cells would be likely to have different effects and could appear similar or opposite to *dSERT* phenotypes respectively.

Our work demonstrates that the increase in overall sleep in *dSERT*$^{16}$ mutants is mediated by at least two changes in sleep architecture that differ between daytime versus nighttime periods. During the nighttime, the increase in sleep in *dSERT*$^{16}$ flies is characterized by significantly decreased bout number (consolidated sleep) whereas during daytime, sleep bout number is increased. In addition, mated *dSERT* females showed an increase in bout frequency compared to controls but only during the daytime and this effect was not observed in males or virgin females. Sex-peptide also induces changes in daytime sleep in mated females [61] and it is possible that sex-peptide and the post-mating response could modify the effects of *dSERT*.

The differential rescue of nighttime or daytime sleep via *TPH-Gal4* or *TRH-Gal4* respectively, supports the idea that different serotonergic circuits may regulate nighttime versus daytime sleep. Previous work has shown that knockdown of *TRH* in the dorsal paired median (DPM) neurons that innervate the MB causes a decrease in nighttime sleep [21]. While the effects on daytime sleep were not clear, the *d5-HT1A* receptor mutant exhibited a clear decrease in nighttime sleep, nighttime bout length, and an increase in nighttime bout number, and all of these behaviors can be genetically rescued via expression of a *d5-HT1A* transgene in the MBs [18]. *TPH-GAL4* but not *TRH-Gal4* driven expression of *UAS-dSERT* rescued the nighttime sleep phenotype of *dSERT*$^{16}$, and *TPH-GAL4* may show higher expression in the MBs than the *TRH-Gal4* driver used in our study [20](S5A–S5B Fig). We speculate that the increased and consolidated nighttime sleep seen in *dSERT*$^{16}$ may be caused by an increase in extracellular serotonin in the MBs and a resultant increase in the activation of 5-HT1A in the MBs; however, further experiments will be needed to directly test this hypothesis.

We also speculate that some aspects of the daytime *dSERT* sleep phenotype, such as the increased frequency of daytime sleep bouts, may be mediated by the ellipsoid body and the d5-HT7 receptor. Activation of EB neurons leads to an increase in the number of sleep episodes specifically during the daytime [20]. *d5-HT7* mutants and treatment with a 5-HT7 antagonist similarly causes a reduction in sleep episode frequency [20]. However, it is likely that the

daytime *dSERT* sleep phenotype is also dependent on other circuits and/or receptors, since *dSERT[16]* exhibits an increase in sleep amount that was not detected in the *d5-HT7* mutant [20].

Serotonin has been previously shown to affect development in both mammals (80,81) and *D. melanogaster* [82–91]. Interestingly, patients with mutations in the autism risk gene *CHD8* show defects in sleep initiation and maintenance, and mutation of the *D. melanogaster* ortholog *kismet* disrupts sleep architecture and increases serotonin labeling of the gut during development [92]. The hyper serotonergic state observed during development appeared to decrease rather than increase sleep [92] in contrast to most of the reported effects of serotonin in the adult fly. It is possible that developmental effects of *dSERT* influence the adult sleep phenotype, and in future experiments we will use genetic rescue during development versus adulthood to explore this idea. For now, we emphasize that we observe an increase in sleep comparable to the phenotype seen in *dSERT[16]* following administration of the SSRI citalopram to wildtype adults. Occlusion of the drug's effects on sleep by the *dSERT[16]* mutant confirm that they were caused by inhibition of dSERT rather than a non-specific target (Fig 3B). These data indicate that inhibition of dSERT in the adult is sufficient to cause at least some aspects of the sleep phenotype, but do not rule out the possibility of additional, or even contrary, developmental effects.

Studies in mice underscore the complexity of serotonin's effects on sleep. For example, an increase in serotonergic signaling caused by increased activity of raphe serotonergic cells reduces REM sleep [93–95], while an increase in serotonin levels caused by a *SERT KO* mouse has the opposite effects and increases REM [96,97]. Knockouts of either 5-HT1B or 5-HT1A also exhibit more REM [98,99], but the increase in REM caused by the SERT KO can be reduced by developmental blockade of 5-HT1A [97].

In humans, the complex relationship between sleep, depression, and the effects of SSRIs underscores the importance of understanding the role of SERT in sleep. Variation in the 5' flanking region of the human *SERT* gene have been associated with a change in SERT expression levels *in vitro* [100] that is linked to the efficacy of SSRIs [101] and sleep deprivation [102]. In addition to sleep, SSRIs impact other critical behaviors such as feeding and mating. SSRIs can induce hypophagic effects and interfere with normal feeding behavior [103–106], and sexual dysfunction is a well-documented side effect of SSRI treatment [46,47]. Our work begins to address the competition between the pathways that regulate these effects in the context of *D. melanogaster* sleep, as we demonstrate that loss of *dSERT* induces an increased sleep drive that appears to outweigh the drive for both normal feeding and mating behaviors (Figs 5–6). We speculate that further studies of *dSERT* mutants may help to uncover the relationship between other serotonergic pathways that regulate complex behaviors common to mammals and flies such as aggression and may be used further as a genetic model for the behavioral effects of SSRIs [70,107–111].

## Materials & methods

### Fly strains

Fly stocks were reared on standard cornmeal media (per 1 L $H_2O$: 12 g agar, 29 g Red Star yeast, 71 g cornmeal, 92 g molasses, 16 mL methyl paraben 10% in EtOH, 10 mL propionic acid 50% in $H_2O$) at 25˚C with 60% relative humidity and entrained to a daily 12 h light,12 h dark schedule (12hr:12hr LD). *w[1118]* flies were obtained from Dr. Henrike Scholz. *UAS-dSERT* (BL24464) and *UAS-MCD8::GFP* (BL32185) were obtained from the Bloomington *Drosophila* Stock Center. *TPH-Gal4* was a gift from Dr. Jongkyeong Chung [71] and *TRH-Gal4* was a gift from Dr. Edward Kravitz [70].

## P-element mutagenesis

The *dSERT* alleles were generated by imprecise excision of the *P{XP}d04388* transposon that is inserted 514 bases upstream of the *dSERT* gene on the second chromosome. First, the *P{XP}d04388* line was backcrossed for five generations to the *w^{1118}* line of the Scholz lab to isogenize the genetic background. Second, females of the *P{XP}d04388* line were crossed to *w^{1118}; CyO/+; [Δ2–3; Ki]/TM2* males. Next, *w^{1118}; P{XP}d04388/CYO; [Δ 2–3; Ki]/+* males were mated with *P{XP}d04388* females. In the next generation, 200 single male flies carrying possible deletion of the *P{XP}d04388* over *P{XP}d04388* were crossed to *Df(2R) PX2/CYO* females that carry a genomic lesion including the complete *dSERT* gene. The new *dSERT* alleles complemented the lethality associated with the deficiency. The males of the F1 generation were screened for loss of the P-element insertion, e.g. white-eyed male flies carrying the *CYO* chromosome. 89 lines were established as stocks. The new alleles were analyzed for genomic lesions using PCR. The following primers were used to confirm lesion and amplify DNA for sequencing: 5'-TGACCCACTAAATGCCATGA-3' and 5' CCAGAAAAAGCGAAATCTGC-3'.

## Quantitative Real Time PCR (qRT-PCR)

Total RNA of 30 male flies was isolated using Trizol followed by a DNAse digest for 30 min at 37° C. To synthesize cDNA, the SuperScript II Reverse Transcriptase (Invitrogen) and oligo^{dT} primers were used according to the manufacturer's instructions. For qRT-PCR analysis, the MESA BLUE qPCR MasterMix Plus for SYBR Assay (Eurogentec) with 100 ng cDNA as template was used. Detection was performed using the iCycler iQ5 Multicolour Real-Time PCR Detection System. The data were analyzed using the ΔΔCt method [112]. The NormFinder software [113] was used for control primer selection. The following control primers recognizing *RpLPO* were selected: 5'-CAG CGT GGA AGG CTC AGT A-3' and 5'-CAG GCT GGT ACG GAT GTT CT-3'. For *dSERT*, the following primers recognizing sequences in the third and fourth exon were used: 5'- GTTGCCTCAGCATCTGGAAG-3' and 5'- CAGCCGA-TAATCGTGTTGTA-3'. Data are shown as fold changes in *dSERT* relative to the controls.

## Western blot analysis

To isolate proteins of fly heads (male and female), 500–1000 flies were frozen in liquid nitrogen and heads were separated with a sieve. The heads were pulverized with a sterile pestle and resuspended in buffer containing 10mM NaCl, 25 mM Hepes, pH 7.5, 2mM EDTA and 1x cOmplete™ Mini protease inhibitors (Merck). The suspension was kept on ice for 10 min with gentle mixing followed by centrifugation at 18300 x g at 4°C for 15 min. The pellet, containing larger membranous particles and the bulk of dSERT was resuspended in a buffer containing 10mM NaCl, 25 mM Hepes, pH 7.5, 1x cOmplete™ Mini protease inhibitors (Merck) and 0.5% CHAPS and incubated at 4°C for 10 minutes with gentle mixing. The solution was centrifuged at 3000 x g for 5 min to recover the membrane extract. 20 μg protein of each sample were used for western blotting according to standard procedures. The membrane was blocked in 5% milk and the primary rabbit anti-dSERT antibody (Giang et al., 2011) was used at a dilution of 1: 20 000 with a mouse anti-β-Actin (mAbcam 8224) at 1:10 000 used as a loading control.

## Behavioral analysis

*Sleep*: Sleep was measured as previously described [114]. In brief, 3–5-day old mated female flies (unless otherwise stated) were individually loaded into 65 mm long glass tubes and inserted into *Drosophila* activity monitors to measure locomotor activity (Trikinetics Inc; Waltham MA, USA) and periods of inactivity for at least 5 minutes were classified as sleep. The

sleep duration and architecture were quantified for both daytime (lights on) and nighttime (lights off), Zeitgeber hours 0 to 12, and 13 to 24, respectively. DAM2s were used for single beam monitors and DAM5Ms were used for multibeam monitors (4 independent beams per tube with 9.1 mm adjacent beam spacing). Custom Visual Basic scripts [114] in Microsoft Excel or SCAMP analysis scripts in MATLAB were used to analyze Trikinetics activity records for sleep behaviors.

Arousal thresholds were assayed by fastening Trikinetics activity monitors to microplate adapters on vortexers (VWR). 0.5 g or 1 g of force was used to stimulate flies for 5 s every hour for 24 hours. The unit g is used to express vibration amplitude with 1 g equal to the gravitational constant. Experiments were performed twice, and arousal rates were averaged together.

### SSRI feeding and sleep

For sleep experiments with SSRIs, flies were loaded into tubes that either contained vehicle or varying concentrations of citalopram hydrobromide (Sigma PHR1640) in vehicle (5% sucrose, 1% agar, and 1% blue food dye in $dH_2O$) and allowed to feed for 20 hours before sleep was analyzed. Blue food dye in vehicle was used to ensure that each fly fed during the initial 20 hours.

### Starvation

For starvation and sleep experiments 3–5-day old, mated females were loaded into DAMs tubes with standard food for acclimation. After 1 day of acclimation, baseline sleep was measured for 24 hours (Day 1). On the experimental day (Day 2) flies were transferred to tubes containing 1% agar at ZT7 to be starved for 17 hours. Flies were then transferred back to fresh tubes containing standard food at ZT0 (Day 3) and activity was recorded for 24-hour recovery period.

### Co-housing

For co-housing sleep experiments flies were loaded individually into DAMs tubes and baseline sleep was recorded for 2 days. On the experimental day at ZT0 flies were combined into male-female or female-female pairs (either same genotype or mixed as specified in the text) in new DAMS tubes and sleep was recorded for 24 hours.

### Grooming

For grooming experiments, 3–5-day old, mated females were observed in individual chambers using a Dinolite USB video camera AM7023CT. A single grooming event was scored when a fly rubbed its legs over its head or abdomen or rubbed legs together. For each fly, grooming was analyzed over three separate 2 min periods and quantified as an average number of grooming events per minute. Each 2 min scoring period was initiated by the observation of a grooming event.

### Negative geotaxis

For negative geotaxis experiments, 3 groups of 10 flies 3-5-days old were placed in two empty 28.5 x 95 mm polystyrene vials fastened together with their open ends facing each other to form a continuous 190 mm space. The long axis of the joined tubes was oriented vertically above a lab bench. After gently tapping the vials on the bench, the flies were allowed to climb for 10 s. Each group was scored for the number of flies that climbed above the 8 cm mark (measured from the bottom of the tube facing the lab bench) compared to those remaining

below the 8 cm mark. Each group was tested 5 times (with one minute recovery between each trial), and the average for the five trials was quantified and used for analysis.

## Locomotor activity and circadian rhythmicity

Experiments were performed with 3-5-day old females in *D. melanogaster* activity monitors (Trikinetics Inc; Waltham MA, USA) that were first entrained for 2–3 days in 12hr:12hr LD. For DD analysis, data were analyzed for 3–6 days from the second day in DD. Data analysis was done with Faas software 1.1 (Brandeis Rhythm Package, Michel Boudinot). Rhythmic flies were defined by $\chi^2$ periodogram analysis with the following criteria: power $\geq$20, width $\geq$2 h with selection of 24 hr ± 8 hr. Power and width are the height and width of the periodogram peak, respectively, and give the significance of the calculated period. The average activity while awake (number of activity counts/time awake ± SEM) was calculated over 3-day period in LD or DD conditions. Morning anticipation index was calculated as (activity for 3 hrs before lights-on)/ (activity for 6 hrs before lights-on) and evening anticipation index as (activity for 3 hrs before lights-off)/(activity for 6 hrs before lights off).

## Courtship & copulation assays

Virgin females were aged 3-5-days in groups of ~10 and naïve males were aged individually for 4 days. Single pairs of female and male flies were gently aspirated into a chamber of a custom acrylic device and video recording was started. The percent of males achieving copulation within 1 hour was measured as well as the time elapsed between the introduction into the chamber and the male display of courtship steps including orientation and wing extension. Analysis of courtship behavior was scored in blind.

## Egg laying and hatchability

Five-day old virgin females fed with wet yeast for one day were placed with males (5 females to 10 males) in one bottle for egg laying on molasses plates over 2 days (with removal and replacement of plates every 24 hours). Once molasses plates were removed, they were kept in sealed Tupperware and after 24 hours in 25°C the number of unhatched eggs were manually counted using a dissecting microscope.

## Feeding assays

Groups of 10 flies 3-5-days old were starved for 24 hours on 1% agar medium as a water source. After starvation, flies were transferred to blue food (10% sucrose, 5% active yeast, 1% agar, and 4% blue food dye (McCormick) and allowed to feed for 15 mins. The amount of blue food dye ingested was measured using visual inspecting feeding scores or spectrophotometry of blue dye as previously described [33]. For visual inspection, each fly was scored with 0 defined as no dye visible in the abdomen, 1 as dye is just visible as a small spot in the crop, 2 as crop with dye extends 1/3-2/3 the length of the abdomen and 3, the abdomen is swollen and filled with dye. The average feeding score was quantified by averaging the feeding scores of each fly in the vial. For spectrophotometric quantification of blue dye, 10 flies were homogenized in 400 μl PBS and centrifuged (14,000 rpm for 3 min at ambient temperature). 250 μl of the supernatant was aspirated into a fresh tube, avoiding the pelleted debris and re-centrifuged (speed, 3 min, temperature). 200 μl of the supernatant was loaded into a 96 well microplate for absorbance readings (Biotek Synergy 2 Multi-Mode Plate Reader). The corrected absorbance of the dye was calculated by subtracting the absorbance at 750 nm (outside of blue dye spectrum) from the absorbance at 630 nm (peak of blue dye) as previously described [33].

## Desiccation and starvation assay

3–day old females were isolated and groups of ten were placed in empty vials (for desiccation) or vials containing 1% agar substrate in $dH_2O$ (for starvation). Flies were visually inspected every 6 hours and surviving flies were manually counted. The assay was designated to be finished when the last fly of each genotype was dead. Data is shown as percentage of live flies over time. For both experiments 10 groups of 10 flies were assayed for each genotype.

## Immunohistochemistry

Immunostaining was performed following a standard procedure comprising brain dissection, fixation in 3% glyoxal for 25 min, blocking in PBTG (PBS plus 0.2% Triton X-100, 0.5% BSA and 2% normal goat serum), and primary and secondary antibody staining diluted in PBTG. The following primary antibodies were used: rabbit anti-GFP (1:500, Invitrogen) and mouse anti-DLG (1:20, 4F3 -Developmental Study Hybridoma Bank). Alexa Fluor 488 and Alexa Flour 555 goat secondary antibody (1:500; Invitrogen) were used as secondary antibodies.

Confocal images were obtained using a Zeiss LSM 880 confocal microscope with Zen software.

Images were processed using Fiji/Image J software.

## Statistics

Data were analyzed in Prism 9 (GraphPad; San Diego CA, USA). Group means were compared using two-tailed t tests or one- or two-way ANOVAs, with repeated-measures where appropriate, and followed by Tukey's post-hoc multiple comparisons tests. Sample sizes for each experiment are depicted in each figure panel or in the appropriate figure legend. All group averages shown in data panels depict mean ± SEM.

## Supporting information

**S1 Fig. Genetic controls, P(doze) and P(wake).** Hourly sleep traces (A) and quantification of total sleep (B) in $w^{1118}$ controls (grey), transheterozygous $dSERT^{10}/dSERT^{16}$ mutants (yellow), homozygous $dSERT^{10}$ mutants (orange), and homozygous $dSERT^{16}$ mutants (red). Hourly sleep traces (C) and quantification of total sleep (D) in $w^{1118}$ controls (grey), $dSERT^{16}$ heterozygotes (purple), and $dSERT^{16}$ homozygous mutants (red). Hourly sleep traces (E) and quantification of total sleep (F) in $w^{1118}$ controls (grey), $dSERT^4$ revertant (light blue), $d04388$ parental, $P$ element line (teal), and $dSERT^{16}$ mutants (red). (G-H) Quantification of P(Wake) (G) and P(Doze) (H) during the light period (LP, Zeitgeber hours 1–12) and dark period (DP, Zeitgeber hours 13–24) in $w^{1118}$ controls (grey), $dSERT^{10}$ mutants (orange), and $dSERT^{16}$ mutants (red). For all panels, sleep traces show mean ± SEM and histograms show both individual datapoints and group means ± SEM. One way ANOVA with Tukey post-hoc test ($p \leq 0.0332^*$, $p \leq 0.0021^{**}$, $p \leq 0.0002^{***}$, $p \leq 0.0001^{****}$). LP and DP were analyzed separately in G, H.
(PDF)

**S2 Fig. $dSERT^{16}$ mutant sleep phenotype is not an artifact of additional amine-linked behaviors.** (A) $dSERT^{16}$ mutants (purple) show no change in grooming behavior compared to $w^{1118}$ controls (grey). The average number of grooming events per minute for three separate 2 min periods with three experimental replicates is shown. In each experimental replicate n = 5 flies for each genotype. Mean± SEM, unpaired Student's t-test. (B) Male and female $dSERT^{16}$ mutants behave indistinguishably from control flies in negative geotaxis assays. Mean ± SEM,

one-way ANOVA.
(PDF)

**S3 Fig. Sleep in *dSERT* mutants of different genders and mating status.** (A) Quantification of total sleep in $w^{1118}$ and $dSERT^{16}$ males (blue), virgin females (pink), or mated females (red). Analysis of daytime sleep (B) and daytime bout frequency (C) and latency (D). Quantification of nighttime sleep (E) and nighttime bout number (F). Graphs show individual datapoints and group means ± SEM. Two-way ANOVA, with Tukey post-hoc test (p≤0.0332*, p≤0.0002***, p≤0.0001****).
(PDF)

**S4 Fig. Arousal threshold of *dSERT^16* flies.** Proportion of $w^{1118}$ (black) and $dSERT^{16}$ (red) flies awakened after 5 seconds of either 0.5 g or 1 g vibrational stimulation during the daytime (A) or nighttime (B) (n = 24 trials per groups, n = 30–32 flies per group). Mean± SEM, Two-way ANOVA with Tukey post-hoc test (≤0.0021**).
(PDF)

**S5 Fig.** (A-B) Representative pictures show expression of UAS-MCD8::GFP (green) driven by *TRH-Gal4* (A) or "*TPH*"-*Gal4* (B) and labeled with an antibody to DLG (magenta) in mushroom bodies. Mushroom body lobes are labeled with white text.
(PDF)

**S1 Data. Raw data.**
(ZIP)

## Acknowledgments

We thank Drs Jongkyeong Chung and Edward Kravitz for sharing fly lines. We thank Dr. Ceazar Nave and Prabhjit Singh in J.D.'s laboratory for technical support and discussion. We thank Dr. Tim Lebestky and Mikhayla Armstrong for their early contributions to the analysis of the *dSERT* mutants.

## Author Contributions

**Conceptualization:** Elizabeth M. Knapp, Rebecca C. Arnold, Maureen M. Sampson, Li Xu, Henrike Scholz, Jeffrey M. Donlea, David E. Krantz.

**Formal analysis:** Elizabeth M. Knapp, Andrea Kaiser, Rebecca C. Arnold, Manuela Ruppert, Li Xu, Matthew I. Anderson, Shivan L. Bonanno, Henrike Scholz, Jeffrey M. Donlea.

**Funding acquisition:** Henrike Scholz, Jeffrey M. Donlea, David E. Krantz.

**Investigation:** Elizabeth M. Knapp, Andrea Kaiser, Rebecca C. Arnold, Maureen M. Sampson, Manuela Ruppert, Li Xu, Matthew I. Anderson, Shivan L. Bonanno, Henrike Scholz, Jeffrey M. Donlea.

**Methodology:** Henrike Scholz, Jeffrey M. Donlea, David E. Krantz.

**Project administration:** Henrike Scholz, Jeffrey M. Donlea, David E. Krantz.

**Resources:** Henrike Scholz, Jeffrey M. Donlea, David E. Krantz.

**Supervision:** Maureen M. Sampson, Henrike Scholz, David E. Krantz.

**Visualization:** Elizabeth M. Knapp, Andrea Kaiser, Rebecca C. Arnold, Manuela Ruppert, Li Xu, Matthew I. Anderson, Shivan L. Bonanno, Henrike Scholz, David E. Krantz.

**Writing – original draft:** Elizabeth M. Knapp.

**Writing – review & editing:** Elizabeth M. Knapp, Li Xu, Henrike Scholz, Jeffrey M. Donlea, David E. Krantz.

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
