## [Decision Letter · Decision Letter 0]

25 Jul 2022

Dear Dr Krantz,

Thank you very much for submitting your Research Article entitled 'Mutation of the Drosophila serotonin transporter dSERT disrupts courtship and feeding and increases both daytime and nighttime sleep' to PLOS Genetics.

The manuscript was fully evaluated at the editorial level and by independent peer reviewers. The reviewers appreciated the attention to an important problem, but raised some substantial concerns about the current manuscript. Based on the reviews, we will not be able to accept this version of the manuscript, but we would be willing to review a much-revised version. We cannot, of course, promise publication at that time.

If you decide to revise the manuscript for further consideration at PLOS Genetics, please aim to resubmit within the next 60 days, unless it will take extra time to address the concerns of the reviewers, in which case we would appreciate an expected resubmission date by email to plosgenetics@plos.org.

[LINK]

We are sorry that we cannot be more positive about your manuscript at this stage. Please do not hesitate to contact us if you have any concerns or questions.

Yours sincerely,

Leslie Griffith

Guest Editor

PLOS Genetics

Gregory P. Copenhaver

Editor-in-Chief

PLOS Genetics

Reviewer's Responses to Questions

**Comments to the Authors:**

Reviewer #1: In this manuscript, Elizabeth and colleagues reported the sleep phenotype of serotonin transporter (SerT) mutant in Drosophila melanogaster. The mutant showed dramatic increase in day and night sleep and also had defects in feeding and copulation. In general, this study used simple evidence to demonstrate multiple behavioral changes on the SerT mutant. There are some suggestions to refine their works.

Major concerns:

1. As we know that, flies have sexual dimorphism in sleep behavior. The authors monitored and analysis the sleep properties of female flies with wild-type or SerT mutant genotype. Did the authors use virgin females? How about the males? The authors should also describe the gender used in RNA and protein experiments.

2. In co-housing experiments, the authors put one male and one female flies in a tube to monitor their sleep. However, only one male or female was monitored as control group. To exclude the possibility that two flies disrupt each other instead of copulation, or the monitoring deviation, the author should put two male or female flies in a tube as control.

3. The author presents multiple behaviors in SerT mutant flies. They showed reduced food intake and didn’t have the effect of starvation induced sleep loss. It may due to the possibility that sleep increase reduce the foraging time. They also showed reduced courtship and copulation behaviors. It is likely that the waking threshold was elevated in SerT mutant. I would suggest that it will be better to make waking threshold tests, which will be easy to explain those changing in distinct behaviors.

4. The author used two gal4 lines, TRH-gal4 and TPH-gal4, to rescue the sleep phenotype caused by SerT mutant. Interestingly, the two gal4 lines rescued different aspect of sleep properties of SerT mutant. I am curious about whether the combination of the two gal4s could rescue the all the defects caused by SerT mutant. If it’s not easy to manipulate, how about a broader gal4, such as a pan-neural gal4? Did the activation or silencing of neurons using TRH-gal4 or TPH-gal4 caused similar sleep phenotype of SerT mutant, respectively?

5. The authors mentioned frequently that dSERT mutant akin to the clinical side effects of SSRIs, they should design experiments, such as using drugs to alter circular 5HT, or the authors should minor the descriptions. Particularly, the authors demonstrated in Fig.6, overexpression of dSERT in TRH-GAL4, flies showed no difference compared with control.

Minor concerns:

1. Please carefully recheck the font in the text (such as line 228), misplace in the figures (such as Figure 3) and mistakes in the methods (such as H20, NaCL, “I will add food recipe”).

2. The behavior of flies, especially sleep, is sensitive to the genetic background. The author backcrossed the P-element line to w1118. However, the cross to [Δ2-3] line and the imprecise excision may also change the genomic background. The author should backcross the flies after the generation of SerT10 and SerT16 lines.

3. Serotonin is involved in development of Drosophila. The author should exclude the development effects on sleep phenotype.

4. For the scatter plot, such as Fig. 2D, Fig. S2A, Fig. 5B, the replicates are not match the number in the brackets, the authors should explain it.

5. For Sup Figure 3C, F, it is more helpful to show whether the mRNA level changed under individual condition other than expression pattern of the TRH-GAL4 and TpH-GAL4. And the expression pattern of TRH-gal4 was so different from previous study (Yongjun Qian, et al., 2017)

Reviewer #2: Knapp et al submitted a study entitled “Mutation of the Drosophila serotonin transporter dSERT disrupts courtship and feeding and increases both daytime and nighttime sleep”. The SERT is the target of most current drugs used to treat depression, and the authors attempt to understand behavioral influences of the inhibition of SERT activity. In this study, authors generated several serotonin transporter mutations of dSERT, and examined several behaviors including sleep, circadian rhythm, feeding and courtship. Interestingly, they found that dSERT mutants exhibit increased daytime and nighttime sleep but opposite changes of sleep architecture during the day and at night. With rescues of dSERT level in two sets of serotonergic neurons, increased daytime or nighttime sleep recovered back to the control level partially. Authors also observed defects of dSERT muants in copulation and food intake. This study may help to further understand how serotonin regulates sleep along with other behaviors. However, there are still many issues to be addressed before consideration of publication at Plos Genetics.

Major issues:

1. P(wake)/P(doze) analysis is based on sleep measurement, it helps to understand the arousal threshold and sleep pressure. Thus, the logic for Figure 1 and 2 should be rearranged. In addition, dSERT mutants exhibited higher P(doze) and lower P(wake) than control flies for both day and night, but cannot interpret the differences of sleep structure during the day and at night. No further evidence to better support and illustrate what results in the opposite phenotypes of day and night sleep architecture.

2. Authors attempted to link sexual interaction with sleep by comparing a pair of male and female to a single fly in the DAM system. Since there are other differences such as social effectors or mechanic influences between a single fly and a pair of flies, the conclusion is relatively weak. Would be more convincing to compare a pair of female and male to a pair of females/males.

3. Food intake was evaluated by starvation of control flies and mutants with the same length of time. As a basic knowledge, the resistance to starvation of each genotype is very different. With the same length of starvation, it possibly results in different starvation status for food intake. Furthermore, it is known that starvation suppresses sleep, but whether less sleep reduction of dSERT mutants is attributed to sleep defects or due to their more resistance to starvation is not conclusive in the present study.

4. dSERT rescue in two sets of GAL4+ neurons resulted daytime or nighttime specific rescue is interesting. Whether dSERT mutation or application of SSRI can rescue insomnia phenotype, and further rescue in a time specific manner is not conclusive here.

Minor issues:

1. In the present study, main findings focused on dSERT on sleep accompanied with observations on feeding and courtship behaviors. The title did not properly reflect the main findings. According to this title, one would expect their main findings on the mechanisms of interplay between feeding and courtship behaviors and sleep.

2. Authors used both single-beam and multibeam monitors to assay whether mutation of dSERT results in changes of activity. There is no description of which beam or how they analyzed the data using the multibeam monitors.

3. The locomotion differences would be better to be evaluated by the activity while awake/the activity while actively moving or using the video tracking analysis.

4. The font and standard need to be consistent. A space is needed between the number and the scale/unit. See: Line97, 350, 352, 390, 393, 402, 431, 432, 441-446. Font issue: Line228-229 and Line373-385, used Arial, not consistent through the entire paper; Line422, description of age of flies are not uniformed. Others: i.e. “H2O, 25 C”, Line491, “MH107390, (DEK),”

5. Many typos: Line 117, “sleep stake”? Line181, “pairing male and females”; Line 245, what does “for” here mean?

6. Citation format is not consistent in the text. i.e. Line 329 “(62)(63)(64)”.

Reviewer #3: In this manuscript, Knapp et al describe the phenotype of two newly created dSERT alleles.

Some aspects of the work are sound, some others need a bit of work. Overall, the work provides novelty in terms of reagents which are certainly going to be very useful for the field. There is less novelty in terms of what it was actually found. Overall I think this can be a useful resource for the field but some further controls are certainly needed.

The main experimental limitation of the entire paper is that the authors missed the opportunity to use the appropriate genetic controls. The process of P-element excision always generates precise or almost-precise excisions that are ideal genetic controls for this type of experiments. The authors report having obtained two of those but did not employ them in any of the comparative studies and opted for the outcrossing background w1118 instead. Given that the rescue experiments did not really give a very convincing rescue, using the appropriate controls is a must, at least for some of the key experiments.

The other (surely addressable) weakness is that some of the experiments are not performed in the optimal way. For instance, Figures 4A,B and 6I use DAM monitors to assess sleep in co-housed flies and that is really just a very approximate way to detect sleep. For this kind of experiments in which you have multiple flies in the same tube you will have to necessarily use videotracking. However, the results in 4C-G are convincing enough so I would simply remove A and B and describe the finding in more anecdotal terms. 4A and B and 6I - if I understood correctly how the experiment was performed - are just not sound enough to be used as "data".

Other suggestions/corrections:

Line 48: Please add an up-to-date reference for “essential for life”.

Line 56: Throughout the paper, please specify Drosophila melanogaster. Drosophila is the name of a genus.

Line 98: This is certainly up to you but I suggest renaming the alleles. Using the apparently random numbers 10, 16, 1 and 4 may be meaningful for you because it probably reflects the order in which they were isolated but it’s uninformative and confusing for the readers.

L462: the food recipe is actually missing

Figure 1C: please do not cut bands from the western blot and show the entire thing, including the molecular weight. This is important for reproducibility.

Figure 4A,B and Figure 6I: One cannot measure sleep in co-housed flies using DAMs. The more flies in the tube, the more IR-breaking beams event the system will record. Remove or redo the experiment using videotracking with multiple fly tracking.

Figure 4C: rather than cap the assay at one hour, it is more informative to measure how long it takes for copulation to occur (that is, latency as you correctly did for D and E).

Figure 5B,C: feeding assays with dyes are good because they are easily and quickly quantifiable with a spectrophotometry of squashed flies (See: J. exp. Biol. 197, 215–235 (1994)). Here you seem to have adopted a fully subjective system, losing most of the strength of the assay. Also, the methods do not specify if the subjective analysis (copulation, feeding etc) were scored in blind.

Giorgio Gilestro

**Have all data underlying the figures and results presented in the manuscript been provided?**

Reviewer #1: None

Reviewer #2: Yes

Reviewer #3: None

PLOS authors have the option to publish the peer review history of their article (what does this mean?). If published, this will include your full peer review and any attached files.

Reviewer #1: No

Reviewer #2: No

Reviewer #3: **Yes: **Giorgio Gilestro

---

## [Decision Letter · Decision Letter 1]

8 Nov 2022

Dear Dr Krantz,

We are pleased to inform you that your manuscript entitled "Mutation of the Drosophila melanogaster serotonin transporter dSERT impacts sleep, courtship, and feeding behaviors" has been editorially accepted for publication in PLOS Genetics. Congratulations!

Yours sincerely,

Leslie Griffith

Guest Editor

PLOS Genetics

Gregory P. Copenhaver

Editor-in-Chief

PLOS Genetics

Comments from the reviewers (if applicable):

Reviewer's Responses to Questions

**Comments to the Authors:**

Reviewer #1: It is now acceptable for publication.

Reviewer #2: Authors made much efforts to strengthen their conclusions and modified their MS substantially, and my concerns have been addressed.

**Have all data underlying the figures and results presented in the manuscript been provided?**

Reviewer #1: None

Reviewer #2: Yes

PLOS authors have the option to publish the peer review history of their article (what does this mean?). If published, this will include your full peer review and any attached files.

Reviewer #1: No

Reviewer #2: No

**Data Deposition**

http://datadryad.org/submit?journalID=pgenetics&manu=PGENETICS-D-22-00678R1

**Press Queries**

---

## [Editor Report · Acceptance letter]

16 Nov 2022

PGENETICS-D-22-00678R1 

Mutation of the Drosophila melanogaster serotonin transporter dSERT impacts sleep, courtship, and feeding behaviors 

Dear Dr Krantz, 

We are pleased to inform you that your manuscript entitled "Mutation of the Drosophila melanogaster serotonin transporter dSERT impacts sleep, courtship, and feeding behaviors" has been formally accepted for publication in PLOS Genetics! Your manuscript is now with our production department and you will be notified of the publication date in due course.

With kind regards,

Livia Horvath

PLOS Genetics

On behalf of:
